# A Hybrid Loss Framework for Decomposition-based Time Series Forecasting Methods: Balancing Global and Component Errors

## Abstract

Accurate time series forecasting, predicting future values based on past data, is crucial for diverse industries. Many current time series methods decompose time series into multiple sub-series, applying different model architectures and training with an end-to-end overall loss for for forecasting. However, this raises a question: does this overall loss prioritize the importance of critical sub-series within the decomposition for the better performance? To investigate this, we conduct a study on the impact of overall loss on existing time series methods with sequence decomposition. Our findings reveal that overall loss may introduce bias in model learning, hindering the learning of the prioritization of more significant sub-series and limiting the forecasting performance. To address this, we propose a hybrid loss framework combining the global and component errors. This framework introduces component losses for each sub-series alongside the original overall loss. It employs a dual min-max algorithm to dynamically adjust weights between the overall loss and component losses, and within component losses. This enables the model to achieve better performance of current time series methods by focusing on more critical sub-series while still maintaining a low overall loss. We integrate our loss framework into several time series methods and evaluate the performance on multiple datasets. Results show an average improvement of 0.5-2% over existing methods without any modifications to the model architectures.

## 1 Introduction

Time series analysis is a powerful tool for understanding and forecasting sequential data points typically measured over time. It finds applications across various domains such as climate science (Wu et al., 2023), transportation (Yin et al., 2021), and energy (Qian et al., 2019b), where recognizing patterns and predicting future values are crucial.

Remarkably, deep learning methods have proven highly effective in time series forecasting by providing robust backbones/model architectures like Multilayer Perceptrons (MLPs) (Zhang et al., 2022b; Chen et al., 2023), Transformers (Vaswani, 2017), and even Large Language Models (LLMs) (Jin et al., 2023; OpenAI, 2023), which are adept at learning complex patterns from large datasets (Godahewa et al., 2021). However, besides the improvement of the model architectures, most of these methods also rely on time series decomposition(Cleveland et al., 1990; Qian et al., 2019a) to better capture various features. Among these, sliding-window decomposition is the most common method, which forms the basis of all model architectures discussed previously (Wu et al., 2021; Zhou et al., 2022; Nie et al., 2022; Zeng et al., 2023). It decomposes a raw time series into seasonal and trend sub-series, representing high-frequency feature (detailed changes) and low-frequency feature (overall trend changes), respectively (Faltermeier et al., 2010). However, although many methods utilize these sub-series, they still employ an end-to-end overall loss function. This loss function computes the difference between the final combined sub-series and the true series. This raises the question: Does optimizing this overall loss guarantee that the features of each sub-series are equally well-learned? Or, could an optimal overall loss fail to optimize the performance of the decomposition-based deep learning model?

To further investigate this, we conduct additional statistical analysis and case studies. Our statistical findings reveal that deep learning methods employing time series decomposition often exhibit significant discrepancies in losses across different sub-series on various datasets. Specifically, the loss on the seasonal sub-series is frequently one to two times smaller than the loss on the trend sub-series, which represents the overall movement of the time series. This disparity in losses suggests that the worse trend component may lead to substantial deviations in the overall trend of the forecast. We further illustrate this issue with a detailed case study.

To address this challenge, we propose a hybrid loss framework combining the global (the overall loss) and component error (the sub-series losses). Inspired by the principles of distributionally robust optimization (DRO) (Wiesemann et al., 2014; Namkoong & Duchi, 2016; Duchi & Namkoong, 2019), we formulate this loss framework as a dual min-max problem. First, we construct a global min-max problem to balance the overall loss and the losses across all sub-series, ensuring that while minimizing the overall loss, the model also dynamically attends to the overall sub-series loss. Furthermore, recognizing that the overall sub-series loss is composed of individual sub-series losses, we formulate a second min-max problem to encourage the model to dynamically focus on potentially higher-loss components during training, thus prioritizing the optimization of critical components like the trend sub-series. We evaluate our loss framework on multiple datasets using existing model architectures and demonstrate an average performance improvement of 0.5-2% without requiring any modifications to the underlying model structures.

In this paper, we make the following contributions:

- Our investigation reveals that the end-to-end overall loss function commonly used in deep learning for time series forecasting may not lead to optimal model performance. Sub-series critical to the overall forecasting might not be sufficiently optimized under the overall loss.

- We propose a novel hybrid loss framework that balances global and component errors to improve time series forecasting by dual min-max.

- The experiments demonstrate the effectiveness of our loss across diverse time series datasets, varying in both length and size, as well as across different models.

## 2 PRELIMINARY EXPERIMENTS

In this section, we explore a potential unifying issue among various deep learning approaches employing time series decomposition when trained under the current loss function. We illustrate this issue through statistical analysis of an experiment and by presenting several intuitive cases.

**Experiment Settings.** To investigate potential shortcomings of existing methods, we reproduce these decomposition-based deep learning methods and, beyond evaluating their overall performance, specifically analyze their performance on each decomposed sub-series. In our experiments,

- For methods, we select DLinear (Zeng et al., 2023), FEDformer (Zhou et al., 2022), and PatchTST (Nie et al., 2022) as representative methods. These methods all employ sliding-window-based time series decomposition (decompose to Seasonal sub-series and Trend sub-series), differing primarily in their backbone architectures: DLinear uses the MLP, while FEDformer and PatchTST utilize transformers. We employ the original loss function of these methods, which computes the Mean Squared Error (MSE) between the combined forecasting of the decomposed sub-series and the ground truth. Notably, these methods also represent the current state-of-the-art in time series forecasting in many benchmarks (Woo et al., 2022; Wang et al., 2024).

- For datasets, our experiments were conducted on four commonly used benchmark datasets: ETTh1, ETTh2 from ETTh (Zhou et al., 2021a), and ETTm1, ETTm2 from ETTm (Zhou et al., 2021b). All datasets are split into training, validation, and testing sets with the 7:1:2 ratio.

- For metrics, performance is evaluated using the standard metrics of Mean Squared Error (MSE) and Mean Absolute Error (MAE).

We show the results of this experiment in Table 1. With these results, we can find that *for deep learning methods employing sliding-window-based time series decomposition, significant discrepancies*

*in forecasting performance across individual sub-series, when trained under the overall loss, may contribute to mainly inaccuracies in the final combined forecasting.* Across the ETTh2, ETTm1, and ETTm2 datasets, the performance on the Trend sub-series is consistently 2 to 5 times worse than the performance on the Seasonal sub-series for all models. Conversely, on the ETTh1 dataset, the Seasonal sub-series performs approximately 2 times worse than the Trend sub-series. Furthermore, comparing the poorly predicted sub-series to the overall forecasting, it accounts for roughly 80% of the overall error. This indicates that a overall loss may not ensure consistent predictive performance across individual sub-series for these decomposition-based methods, suggesting a biased learning towards certain components of sub-series. More importantly, this bias appears to be a major contributor to the overall forecasting error of these methods.

Table 1: Multivariate time series forecasting results on four datasets with sliding-window-based deep learning methods. The results are based on the average of prediction lengths {96, 192, 336, 720} with input length 96. A lower MSE and MAE indicates better performance. The "Global/Components" column indicates whether the reported results represent the overall forecasting performance or the performance on each individual decomposed sub-series.

| Models | Global/Componets | ETTh1 | | ETTh2 | | ETTm1 | | ETTm2 | |
|---|---|---|---|---|---|---|---|---|---|
| | | MSE | MAE | MSE | MAE | MSE | MAE | MSE | MAE |
| Dlinear | Overall | 0.4588 | 0.4519 | 0.4981 | 0.4792 | 0.4061 | 0.4102 | 0.3102 | 0.3670 |
| | Seasonal | 0.2965 | 0.3604 | 0.0888 | 0.2071 | 0.0969 | 0.2115 | 0.0486 | 0.1467 |
| | Trend | 0.1716 | 0.3146 | 0.4144 | 0.4264 | 0.3192 | 0.3726 | 0.2661 | 0.3371 |
| FEDformer | Overall | 0.4394 | 0.4581 | 0.4429 | 0.4549 | 0.4441 | 0.4543 | 0.3031 | 0.3493 |
| | Seasonal | 0.2793 | 0.3703 | 0.0866 | 0.2102 | 0.1010 | 0.2116 | 0.0446 | 0.1381 |
| | Trend | 0.1678 | 0.3172 | 0.3551 | 0.4048 | 0.3249 | 0.3979 | 0.2595 | 0.3133 |
| Patchtst | Overall | 0.4506 | 0.4411 | 0.3658 | 0.3945 | 0.3838 | 0.3954 | 0.2821 | 0.3261 |
| | Seasonal | 0.3031 | 0.3656 | 0.0835 | 0.1971 | 0.1319 | 0.2378 | 0.0490 | 0.1436 |
| | Trend | 0.1566 | 0.2935 | 0.2710 | 0.3277 | 0.3198 | 0.3641 | 0.2328 | 0.2846 |

To further explore the practical impact of this bias and provide a visual illustration, we conduct the case studies for each method across the different datasets, as shown in Figure 1.[1] We can find that *sub-series with larger losses, especially the Trend sub-series, does have a greater impact on the overall forecasting.* In Figure 1 (a) and (b), the models accurately predict the Seasonal sub-series, but fail to capture the increasing trend in the Trend sub-series. This leads to a visually apparent underestimation of the overall forecasting compared to the ground truth. In contrast, for Figure 1 (c), the primary error occurs in the middle-early part, where both sub-series have significant errors. Although the Seasonal sub-series exhibits larger errors in the later part, the more accurate Trend sub-series forecasting results also can make a smaller overall error. Therefore, this further confirms that the overall loss may not effectively optimize for the sub-series that contribute significantly to the overall forecasting.

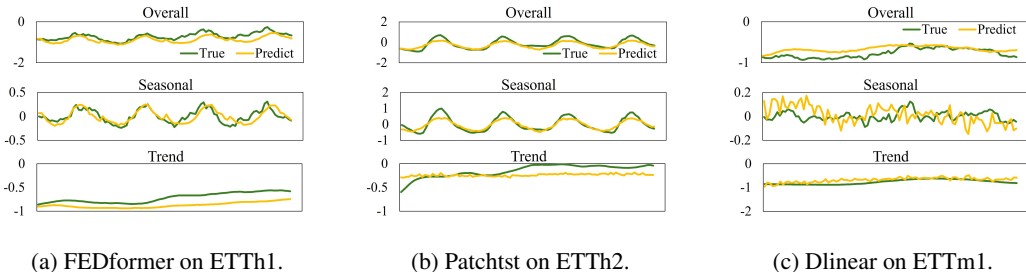

(a) FEDformer on ETTh1.    (b) Patchtst on ETTh2.    (c) Dlinear on ETTm1.

Figure 1: The case study of time series forecasting. The results show the prediction-length-96 part (input length is 96) for different methods on different datasets. Each sub figure presents the single-variate (last variate) overall forecasting part and the forecasting part of the individual sub-series.

---

[1]More cases can be found in Appendix D.

## 3 METHOD

As revealed in the previous section, a overall loss indeed introduces bias when training deep learning methods on decomposed sub-series, potentially leading to significant errors, particularly in the Trend sub-series. To address this issue, we propose a hybrid loss framework in this section, which incorporates component-specific (sub-series) losses alongside the overall loss, and dynamically adjusts their weights to improve the overall and sub-series forecasting.

Specifically, we define the overall loss as $Loss_G$ (compute the MSE on the final results), the component loss as $Loss_C$, which is the sum of $Loss_S$ and $Loss_T$ for Seasonal and Trend sub-series (compute the MSE on the Seasonal sub-series and Trend sub-series results), respectively. We use a dual min-max problem to first balance the losses of $Loss_G$ and $Loss_C$, and then balance the losses of $Loss_S$ and $Loss_T$. This aims to maintain the overall forecasting performance while also focusing on and dynamically balancing the forecasting of sub-series with larger errors.

### 3.1 OPTIMIZATION OBJECTIVE

Drawing inspiration from distributionally robust optimization (DRO), our previous goal is to achieve optimal forecasting for the max loss part in our hybrid loss framework by adjusting the losses of $Loss_G$ and $Loss_C$, and the losses of $Loss_S$ and $Loss_T$ through dual min-max weighting. We define the first (min-max) optimization objective as follows:

$$\min_{\theta} \max_{w_1+w_2=1, w_i \geq 0} w_1 Loss_G + w_2 Loss_C, \tag{1}$$

where $\theta$ means the parameters of the deep learning method, $w_1$ and $w_2$ mean the weights for the overall loss $Loss_G$ and the component loss $Loss_C$ respectively, and the $Loss_C$ is associated with the second (min-max) optimization objective:

$$Loss_C = \min_{\theta} \max_{\alpha+\beta=1, \alpha, \beta \geq 0} \alpha Loss_S + \beta Loss_T, \tag{2}$$

where $\alpha$ and $\beta$ mean the weights for the Seasonal loss $Loss_S$ and the Trend loss $Loss_T$ respectively.

Therefore, the Equation 1 means that when the component loss exceeds the overall loss, we need the model to prioritize the forecasting performance of sub-series rather than solely focusing on the final forecasting, and the Equation 2 means that when optimizing for sub-series, we need the model to prioritize these with larger losses, as they are often the primary contributors to errors in the overall forecasting. We can combine these two optimization objectives as follows:

$$\min_{\theta} \max_{\substack{w_1+w_2=1, w_i \geq 0 \\ \alpha+\beta=1, \alpha, \beta \geq 0}} w_1 Loss_G + w_2(\alpha Loss_S + \beta Loss_T). \tag{3}$$

### 3.2 IMPLEMENTATION

To solve this optimization problem Equation 3, we also need to optimize the parameters $w_1$, $w_2$, $\alpha$ and $\beta$. Instead of applying the gradient descent method, we use estimation technique as the mirror descent method from DRO (Zhang et al., 2022a) to update as follows:

$$w_1^{cur} = \frac{w_1^{pre} \exp(\lambda_1 Loss_G)}{w_1^{pre} \exp(\lambda_1 Loss_G) + w_2^{pre} \exp(\lambda_1 Loss_C)}, \tag{4}$$

$$w_2^{cur} = \frac{w_2^{pre} \exp(\lambda_1 Loss_C)}{w_1 \exp(\lambda_1 Loss_G) + w_2^{pre} \exp(\lambda_1 Loss_C)}, \tag{5}$$

$$\alpha^{cur} = \frac{\alpha^{pre} \exp(\lambda_2 Loss_S)}{\alpha^{pre} \exp(\lambda_2 Loss_S) + \beta^{pre} \exp(\lambda_2 Loss_T)}, \tag{6}$$

$$\beta^{cur} = \frac{\beta^{pre} \exp(\lambda_2 Loss_T)}{\alpha^{pre} \exp(\lambda_2 Loss_S) + \beta^{pre} \exp(\lambda_2 Loss_T)}, \tag{7}$$

where $pre$ denotes the previous update step, $cur$ denotes the current update step. The $\lambda_i$ is a hyperparameter that balances the importance of the weighting term. Its value is often determined by the properties of the deep learning method being used. We initialize $w_1, w_2, \alpha, \beta = \frac{1}{2}$ in the initial iteration of our experiments.

The optimization process then becomes: for each optimization step, we first compute the weights of the individual losses using the equations above, resulting in the combined loss

$$Loss = w_1 Loss_G + w_2(\alpha Loss_S + \beta Loss_T), \tag{8}$$

which is then used to update the model parameters $\theta$ [2].

Table 2: Multivariate time series forecasting results on deep learning methods with/without hybrid loss framework. The "Loss" indicates what kind of the loss does the methods use.

| Models | | Dlinear | | FEDformer | | Patchtst | |
|---|---|---|---|---|---|---|---|
| Datasets | Loss | Original | Hybrid Loss | Original | Hybrid Loss | Original | Hybrid Loss |
| ETTh1 | MSE | 0.4588 | **0.4579** | 0.4394 | **0.4380** | 0.4506 | **0.4502** |
| | MAE | 0.4519 | **0.4511** | 0.4581 | **0.4573** | 0.4411 | **0.4402** |
| ETTh2 | MSE | 0.4981 | **0.4974** | 0.4429 | **0.4417** | 0.3658 | **0.3639** |
| | MAE | 0.4792 | **0.4785** | 0.4549 | **0.4539** | 0.3945 | **0.3929** |
| ETTm1 | MSE | 0.4061 | **0.4060** | 0.4441 | **0.4424** | 0.3838 | **0.3813** |
| | MAE | **0.4102** | **0.4102** | 0.4543 | **0.4535** | 0.3954 | **0.3943** |
| ETTm2 | MSE | 0.3102 | **0.3100** | 0.3031 | **0.3021** | 0.2821 | **0.2790** |
| | MAE | 0.3670 | **0.3667** | 0.3493 | **0.3480** | 0.3261 | **0.3247** |
| Electricity | MSE | 0.2095 | **0.2093** | **0.2141** | 0.2224 | **0.1951** | 0.1955 |
| | MAE | 0.2956 | **0.2955** | **0.3261** | 0.3334 | **0.2794** | 0.2796 |
| Exchange | MSE | 0.3357 | **0.3307** | **0.5017** | 0.5201 | **0.3517** | 0.3531 |
| | MAE | 0.3948 | **0.3947** | **0.4908** | 0.5025 | **0.3963** | 0.3966 |
| illness | MSE | 2.3465 | **2.3452** | 2.7893 | **2.4759** | 1.6318 | **1.5197** |
| | MAE | **1.0883** | 1.0892 | 1.1200 | **1.0974** | 0.8616 | **0.8279** |
| Weather | MSE | 0.2670 | **0.2638** | 0.3128 | **0.3112** | 0.2598 | 0.2605 |
| | MAE | 0.3174 | **0.3076** | 0.3609 | **0.3589** | 0.2816 | **0.2798** |

## 4 EXPERIMENT

In this section, we aim to validate the effectiveness of our proposed hybrid loss framework for both overall and sub-series forecasting performance across multiple datasets. We also conduct the ablation studies to analyze the contribution of each component of our loss framework.

### 4.1 EXPERIMENTAL SETUP

**Datasets.** For the time series forecasting tasks, in addition to ETTh1, ETTh2, ETTm1, and ETTm2 datasets used in our preliminary experiments, we incorporate 4 more commonly used datasets: Electricity (Trindade, 2015), Exchange-rate (Exchange) (Lai et al., 2018), National-illness (illness) (Zhou et al., 2021b), and Weather[3], to demonstrate the broader applicability of our loss framework. These 4 datasets split into training, validation, and testing sets with the 3:1:1 ratio.

**Baseline.** Given that the models used in our preliminary experiments, DLinear (Zeng et al., 2023), Fedformer (Zhou et al., 2022), and PatchTST (Nie et al., 2022), are already among the most prominent and effective, covering both MLP and transformer backbones, as well as point and patch em-

---

[2]The effectiveness and convergence of this optimization process are supported by prior work (Duchi & Namkoong, 2019).

[3]https://www.bgc-jena.mpg.de/wetter/

bedding variants, we retain these models as baselines. Our method directly replaces the original loss function of these baselines with our proposed hybrid loss framework during training.[4]

**Implementation details.** In our experiments, except the nation-illness dataset, all the input lengths are 96, and prediction lengths are $\{96, 192, 336, 720\}$, respectively. For nation-illness dataset, the input length is 104 and prediction lengths are $\{24, 36, 48, 60\}$, respectively. To conserve space, the results presented in this section are averaged across all these prediction lengths.[5] Based on our validation set performance, we set $\lambda_1 = 0.9$ and $\lambda_2 = 0.1$ for our hybrid loss framework across all models and datasets. All experiments were conducted on a system with two NVIDIA V100 32G GPUs and an Intel(R) Xeon(R) CPU E5-2678 v3 @ 2.50GHz with 128GB of RAM.

**Metrics.** We use the standard metrics of Mean Squared Error (MSE) and Mean Absolute Error (MAE) after the data normalization. A lower MSE and MAE indicates better performance.[6]

### 4.2 MAIN RESULTS

**Our hybrid loss framework effectively improves the final performance of existing methods across a wide range of datasets.** Table 2 presents the overall forecasting performance of these methods using both the original loss and our proposed hybrid loss framework. We observe improvements across most datasets. The magnitude of improvement is generally around 0.5-2%, with a notable exception on the illness dataset where our method boosts the performance of FEDformer by nearly 10% on MSE. This demonstrates that the dynamic focus on sub-series losses introduced by our hybrid loss framework is indeed effective and ultimately leads to improved overall performance.

We further investigate the reasons for the worse performance of FEDformer and PatchTST with our hybrid loss framework on the Electricity and Exchange-rate datasets. We find that the time series in these datasets lack readily discernible patterns and exhibit numerous abrupt changes. Consequently, incorporating sub-series losses reinforces the tendency to learn a smoother, low-frequency representation for each sub-series, which leads to less accurate forecasting in the final results.

Table 3: Multivariate time series forecasting overall and subseries results on deep learning methods with/without hybrid loss framework. The "Global/Components" column indicates whether the reported results represent the overall forecasting performance or the performance on each individual decomposed sub-series. The "Loss" column indicates what kind of the loss does the methods use.

| Models | Loss | Global/Componets | ETTh1 | | ETTh2 | | ETTm1 | | ETTm2 | |
|---|---|---|---|---|---|---|---|---|---|---|
| | | | MSE | MAE | MSE | MAE | MSE | MAE | MSE | MAE |
| Dlinear | Original | Overall | 0.4588 | 0.4519 | 0.4981 | 0.4792 | 0.4061 | 0.4102 | 0.3102 | 0.3670 |
| | | Seasonal | 0.2965 | 0.3604 | 0.0888 | 0.2071 | 0.0969 | 0.2115 | 0.0486 | 0.1467 |
| | | Trend | 0.1716 | 0.3146 | 0.4144 | 0.4264 | 0.3192 | 0.3726 | 0.2661 | 0.3371 |
| | Hybrid | Overall | 0.4579 | 0.4521 | 0.4974 | 0.4785 | 0.4060 | 0.4102 | 0.3100 | 0.3667 |
| | | Seasonal | 0.2923 | 0.3556 | 0.0819 | 0.1959 | 0.0961 | 0.2106 | 0.0435 | 0.1323 |
| | | Trend | 0.1686 | 0.3122 | 0.4038 | 0.4203 | 0.3189 | 0.3724 | 0.2611 | 0.3299 |
| FEDformer | Original | Overall | 0.4394 | 0.4581 | 0.4429 | 0.4549 | 0.4441 | 0.4543 | 0.3031 | 0.3493 |
| | | Seasonal | 0.2793 | 0.3703 | 0.0866 | 0.2102 | 0.1010 | 0.2116 | 0.0446 | 0.1381 |
| | | Trend | 0.1678 | 0.3172 | 0.3551 | 0.4048 | 0.3249 | 0.3979 | 0.2595 | 0.3133 |
| | Hybrid | Overall | 0.4380 | 0.4573 | 0.4417 | 0.4539 | 0.4424 | 0.4535 | 0.3021 | 0.3480 |
| | | Seasonal | 0.2786 | 0.3697 | 0.0826 | 0.2021 | 0.0944 | 0.2038 | 0.0415 | 0.1300 |
| | | Trend | 0.1649 | 0.3144 | 0.3409 | 0.4010 | 0.3187 | 0.3933 | 0.2570 | 0.3095 |
| Patchtst | Original | Overall | 0.4506 | 0.4411 | 0.3658 | 0.3945 | 0.3838 | 0.3954 | 0.2821 | 0.3261 |
| | | Seasonal | 0.3031 | 0.3656 | 0.0835 | 0.1971 | 0.1319 | 0.2378 | 0.0490 | 0.1436 |
| | | Trend | 0.1566 | 0.2935 | 0.2710 | 0.3277 | 0.3198 | 0.3641 | 0.2328 | 0.2846 |
| | Hybrid | Overall | 0.4502 | 0.4402 | 0.3699 | 0.3989 | 0.3813 | 0.3963 | 0.2790 | 0.3247 |
| | | Seasonal | 0.3008 | 0.3643 | 0.1116 | 0.1689 | 0.0880 | 0.1997 | 0.0451 | 0.1361 |
| | | Trend | 0.1521 | 0.2900 | 0.3020 | 0.3250 | 0.2746 | 0.3315 | 0.2304 | 0.2811 |

**Our hybrid loss framework significantly enhances the forecasting performance of individual sub-series, particularly the Trend sub-series.** Since our hybrid loss framework aims to improve

---

[4]We also provide a comparison with a wider range of models in Appendix B.1, demonstrating that models utilizing our hybrid loss framework still achieve state-of-the-art performance in a broader comparison.

[5]More detailed results of each prediction length are provided in Appendix B.2.

[6]More details of this section can be found in Appendix A.

overall performance by enhancing the forecasting of individual sub-series, we conduct additional experiments on the four datasets used in our preliminary experiments to compare the performance of our hybrid loss framework against the original loss, shown in Table 3. These results clearly demonstrate a significant reduction in the forecasting error of individual sub-series when using our hybrid loss framework. This improvement is particularly pronounced for the Trend sub-series, often exceeding a 2% reduction in error. Given that most datasets exhibit greater forecasting deficiencies in the Trend component, and considering the importance of the Trend sub-series in representing the overall series trajectory, we believe our hybrid loss framework effectively addresses a common weakness in current decomposition-based deep learning methods.

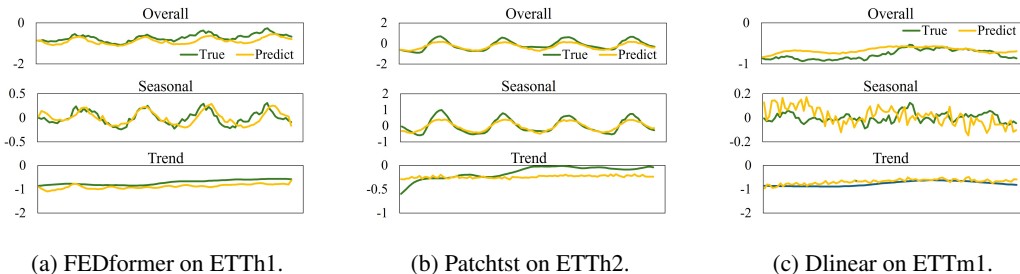

| (a) FEDformer on ETTh1. | (b) Patchtst on ETTh2. | (c) Dlinear on ETTm1. |

Figure 2: The case study of time series forecasting results with our hybrid loss framework. The settings are the same as Figure 1.

We also perform the case study, using the same setup as in the preliminary experiments, to visually analyze the effects of our hybrid loss framework, as shown in Figure 2.[7] Comparing Figure 1 with Figure 2, we observe a clear visual improvement in the forecasting of sub-series, particularly the Trend sub-series in Figure 2 (a) and (b). These now capture the upward trend, leading to better overall forecasting. DLinear on ETTm1 also shows a visually perceptible improvement in overall performance, despite some residual high-frequency errors in the Seasonal sub-series. Therefore, we believe learning the Trend sub-series may be a promising area for future research discovered by this work, and there is still room for further improvement even with the hybrid loss framework.

### 4.3 ABLATION STUDY

To further explore our hybrid loss framework, we conduct two ablation studies: one to analyze the effectiveness of different components/variants, and another to assess the impact of varying initial hyperparameter settings.

Table 4: The ablation study of multivariate time series forecasting results on deep learning methods with our hybrid loss framework or its variants. The "Loss" indicates what kind of the loss does the methods use.

|  |  | ETTh1 | | ETTh2 | | ETTm1 | | ETTm2 | |
| --- | --- | --- | --- | --- | --- | --- | --- | --- | --- |
| Models | Loss | MSE | MAE | MSE | MAE | MSE | MAE | MSE | MAE |
| | Hybrid | **0.4579** | **0.4521** | **0.4974** | **0.4785** | 0.4060 | 0.4102 | **0.3100** | **0.3667** |
| Dlinear | Componet | 0.4593 | 0.4539 | 0.5603 | 0.5150 | **0.3978** | **0.4101** | 0.3239 | 0.3767 |
| | Fix weight | 0.4596 | 0.4533 | 0.4993 | 0.4788 | 0.4079 | 0.4122 | 0.3164 | 0.3678 |
| | Hybrid | **0.4380** | **0.4573** | **0.4417** | **0.4539** | **0.4424** | **0.4535** | **0.3021** | **0.3480** |
| FEDformer | Componet | 0.4879 | 0.4858 | 0.4574 | 0.4651 | 0.4778 | 0.4698 | 0.3251 | 0.3722 |
| | Fix weight | 0.4414 | 0.4600 | 0.4456 | 0.4568 | 0.4670 | 0.4637 | 0.3055 | 0.3511 |
| | Hybrid | **0.4502** | **0.4402** | 0.3699 | 0.3989 | **0.3813** | **0.3963** | **0.2790** | 0.3247 |
| Patchtst | Componet | 0.4620 | 0.4498 | **0.3691** | **0.3966** | 0.3917 | 0.4001 | 0.2801 | **0.3240** |
| | Fix weight | 0.4586 | 0.4871 | 0.3742 | 0.4031 | 0.3900 | 0.3985 | 0.2792 | 0.3267 |

For the first ablation study, we compare two variants of our hybrid loss framework in the datasets used in our preliminary experiments. The first variant, denoted as "Component", uses only the

---

[7]More cases can be found in Appendix D.

sub-series loss, corresponding to loss function Equation 2. The second variant, denoted as "Fixed Weight", uses fixed weights $w_1, w_2, \alpha, \beta$, all set to 0.5, during model training. The results of this ablation study are presented in Table 4.

**Using only the sub-series loss is insufficient, and dynamically updating the weights during training is crucial.** In Table 4, our hybrid loss framework achieves the best performance in most cases, often outperforming the "Component" variant (using only sub-series loss with min-max) by over 3% and the "Fixed Weight" variant by 1-2%. This demonstrates that solely focusing on the sub-series loss is insufficient; while the model may learn to predict sub-series well, the combined forecasting remains inaccurate. Furthermore, it highlights the dynamic nature of the balance between overall and sub-series losses during training, emphasizing that neither the overall loss nor any single sub-series consistently dominates the optimization process.

For the second ablation study, we compare the impact of different initial weights $w_1, w_2, \alpha, \beta$, to explore the influence of initial bias towards specific components of loss. As established in Section 3.1, $w_1 + w_2 = 1$ and $\alpha + \beta = 1$. Therefore, we test the following combinations: $w_1 = 0.1, \alpha = 0.1$ or 0.9; $w_1 = 0.5, \alpha = 0.5$; and $w_1 = 0.9, \alpha = 0.1$ or 0.9. We used all datasets from the preliminary experiments and the DLinear and PatchTST models.[8] The results of this ablation study are presented in Table 5.

**For most datasets, uniform initial weights (0.5) provide good performance, while excessive bias in the initial weights may lead to performance degradation.** As shown in Table 5, drastically altering the initial weights can still impact the final performance. The uniform initial weights (0.5) generally maintain stable performance and, in many cases, outperform initializations with 0.1 or 0.9 by approximately 1%. This suggests that, in the absence of prior knowledge about the data, using a balanced set of initial weights (e.g., 0.5) for the our hybrid loss framework allows the model to learn and adjust these weights during training, leading to more reliable final performance compared to aggressively setting the initial weights.

Table 5: The ablation study of multivariate time series forecasting results on our hybrid loss framework with different initial weights. As $w_1 + w_2 = 1$ and $\alpha + \beta = 1$, we only specify the initial values of $w_1$ and $\alpha$ in the table.

| Models | $w_1$ | $\alpha$ | ETTh1 | | ETTh2 | | ETTm1 | | ETTm2 | |
|---|---|---|---|---|---|---|---|---|---|---|
| | | | MSE | MAE | MSE | MAE | MSE | MAE | MSE | MAE |
| Dlinear | 0.1 | 0.1 | 0.4596 | 0.4524 | 0.4939 | 0.4779 | 0.4091 | 0.4142 | 0.3064 | 0.3638 |
| | | 0.9 | 0.4610 | 0.4541 | 0.4978 | 0.4791 | 0.4047 | 0.4097 | 0.3091 | 0.3658 |
| | 0.5 | 0.5 | 0.4579 | 0.4511 | 0.4974 | 0.4785 | 0.4060 | 0.4102 | 0.3100 | 0.3667 |
| | 0.9 | 0.1 | 0.4589 | 0.4519 | 0.4982 | 0.4793 | 0.4050 | 0.4103 | 0.3102 | 0.3670 |
| | | 0.9 | 0.4589 | 0.4519 | 0.4983 | 0.4793 | 0.4050 | 0.4103 | 0.3102 | 0.3670 |
| Patchtst | 0.1 | 0.1 | 0.4554 | 0.4451 | 0.3651 | 0.3940 | 0.3836 | 0.3975 | 0.2802 | 0.3200 |
| | | 0.9 | 0.4518 | 0.4412 | 0.3650 | 0.3937 | 0.3828 | 0.3973 | 0.2800 | 0.3254 |
| | 0.5 | 0.5 | 0.4502 | 0.4402 | 0.3639 | 0.3929 | 0.3813 | 0.3943 | 0.2790 | 0.3247 |
| | 0.9 | 0.1 | 0.4493 | 0.4393 | 0.3642 | 0.3935 | 0.3821 | 0.3960 | 0.2792 | 0.3247 |
| | | 0.9 | 0.4492 | 0.4391 | 0.3642 | 0.3934 | 0.3821 | 0.3935 | 0.2792 | 0.3248 |

## 5 RELATED WORKS

**The deep learning backbones in time series forecasting.** Deep learning dominates time series forecasting in recent years. These methods leverage different powerful neural network architectures as backbones, adapting them to capture the characteristics of time series and learn effective predictive patterns from large datasets. For example, Prior to the rise of transformers, CNNs (Hewage et al., 2020) and RNNs (Lai et al., 2018) demonstrated the potential of deep learning to surpass traditional forecasting methods. Subsequently, transformers became the prevalent backbone, with models like Informer (Zhou et al., 2021b), Autoformer (Wu et al., 2021), Fedformer (Zhou et al.,

---

[8]We also conduct this ablation study on FEDformer using the ETTh1 and ETTh2 datasets, presented in Appendix C.

2022), iTransformer (Liu et al., 2023), and PatchTST (Nie et al., 2022) specifically designed to exploit the sequential nature of time series. However, recent work suggests that simpler architectures, like MLP-based models such as DLinear (Zeng et al., 2023), TimeMixer (Wang et al., 2024), and TimesNet (Wu et al., 2022), can also achieve comparable or even superior performance. Furthermore, the impressive reasoning and generalization abilities of recent large language models (LLMs) (Jin et al., 2023) have spurred exploration of their potential for zero-shot time series forecasting (Jin et al., 2023; Das et al., 2023). While these backbone architectures constitute the majority of time series forecasting research, many of them still employ time series decomposition techniques to better capture temporal dynamics by learning representations for individual sub-series. Furthermore, these models still rely on end-to-end overall loss functions(Jadon et al., 2024), leaving the relationship between the loss and the effectiveness of the learning of sub-series unexplored.

**The times series decomposition in time series forecasting.** Time series decomposition is a crucial component in many contemporary time series forecasting models, employed across various backbone architectures (Wu et al., 2021; Zhou et al., 2022; Nie et al., 2022; Zeng et al., 2023). Its core principle involves decomposing a raw time series into two or more sub-series, each representing specific characteristics of the original series. For example, the widely used sliding window approach (Faltermeier et al., 2010) decomposes a time series into seasonal and trend components, capturing the local fluctuations and overall trajectory, respectively. Other models explore alternative decomposition methods based on mathematical principles. Fedformer (Zhou et al., 2022) builds upon the sliding window approach by further decomposing sub-series using Fourier transforms, focusing on dominant frequencies. TimeMixer (Wang et al., 2024) utilizes a multi-scale decomposition to capture information at different granularities. In this work, we specifically investigate how to enhance the learning of decomposed sub-series, particularly focusing on the commonly used sliding window decomposition method.

## 6 CONCLUSION

We explore the potential shortcomings of existing deep learning time series forecasting methods that incorporate time series decomposition. We find that the end-to-end overall loss employed by these methods may hinder the effective learning of decomposed sub-series, ultimately impacting the final performance. Therefore, we propose a novel hybrid loss framework designed to address this balance between different sub-series and the overall series in time series forecasting. By employing a dual min-max loss framework, our approach dynamically emphasizes both the overall series and the sub-series that require enhanced learning. This avoids the bias that occurs when focusing solely on overall loss, which may lead to suboptimal model performance. Our framework achieves state-of-the-art performance across a wide range of datasets and experiments demonstrate that this loss framework can yield an average improvement of 0.5-2% across existing time series models.

### LIMITATIONS AND FUTURE WORK

Despite the work presented in this study, from problem identification to solution development for decomposition-based deep learning methods in time series forecasting, our work has some limitations that we hope to address in future work.

First, although the investigated time series forecasting methods represent the current state-of-the-art, they all rely on sliding window decomposition. While alternative decomposition methods are less common, their performance under our loss framework may also need further investigation.

Second, due to computational constraints associated with averaging results across multiple prediction lengths for each datasets, the datasets used in our preliminary and ablation experiments could be expanded further to provide more comprehensive validation of our loss framework's effectiveness.

### REPRODUCIBILITY

The code for our hybird loss framework is available in the Supplementary Material we submitted. It is designed as a plug-and-play module readily applicable to existing time series forecasting methods utilizing sliding window decomposition.

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

# A    MORE DETAILS

We show more details of datasets, evaluation metrics, experiments in this section.

**Datasets details.** We evaluate the performance the methods on 8 commonly used datasets: ETTh1 (Zhou et al., 2021a), ETTh2 (Zhou et al., 2021a), ETTm1 (Zhou et al., 2021b), ETTm2 (Zhou et al., 2021b), Electricity (Trindade, 2015), Exchange-rate (Exchange) (Lai et al., 2018), National-illness (illness) (Zhou et al., 2021b), and Weather[9]. Following the standard settings of the existing benchmarks (Zeng et al., 2023; Zhou et al., 2022; Nie et al., 2022), except the nation-illness dataset, all the input lengths are 96, and prediction lengths are {96, 192, 336, 720}, respectively. For nation-illness dataset, the input length is 104 and prediction lengths are {24, 36, 48, 60}, respectively. The first 4 datasets split into training, validation, and testing sets with the 7:1:2 ratio, and the last 4 datasets split into training, validation, and testing sets with the 3:1:1 ratio. The detailed descriptions of these datasets in Table 6.

Table 6: Dataset detailed descriptions. The dataset size is organized in (Train, Validation, Test). The "Dim" means the dimensions of the multivariate in the dataset.

| Dataset | Dim | Input Length | Prediction Length | Dataset Size | Frequency | Information |
|---|---|---|---|---|---|---|
| ETTh1 | 7 | 96 | 96 | (8449, 2785, 2785) | Hourly | Temperature |
| | | | 192 | (8353, 2689, 2689) | | |
| | | | 336 | (8209, 2545, 2545) | | |
| | | | 720 | (7825, 2161, 2161) | | |
| ETTh2 | 7 | 96 | 96 | (8449, 2785, 2785) | Hourly | Temperature |
| | | | 192 | (8353, 2689, 2689) | | |
| | | | 336 | (8209, 2545, 2545) | | |
| | | | 720 | (7825, 2161, 2161) | | |
| ETTm1 | 7 | 96 | 96 | (34369, 11425, 11425) | 15 mins | Temperature |
| | | | 192 | (34273, 11329, 11329) | | |
| | | | 336 | (34129, 11185, 11185) | | |
| | | | 720 | (33745, 10801, 10801) | | |
| ETTm2 | 7 | 96 | 96 | (34369, 11425, 11425) | 15 mins | Temperature |
| | | | 192 | (34273, 11329, 11329) | | |
| | | | 336 | (34129, 11185, 11185) | | |
| | | | 720 | (33745, 10801, 10801) | | |
| Electricity | 321 | 96 | 96 | (18221, 2537, 5165) | Hourly | Electricity |
| | | | 192 | (18125, 2441, 5069) | | |
| | | | 336 | (17981 , 2297, 4925) | | |
| | | | 720 | (17597, 1913, 4541) | | |
| Exchange_rate | 8 | 96 | 96 | (5120, 665, 1422) | Day | Exchange rates |
| | | | 192 | (5024, 569, 1326) | | |
| | | | 336 | (4880, 425, 1182) | | |
| | | | 720 | (4496, 41, 798) | | |
| illness | 7 | 104 | 24 | (549, 74, 170) | Week | National illness |
| | | | 36 | (537, 62, 158) | | |
| | | | 48 | (525, 50, 146) | | |
| | | | 60 | (513, 38, 134) | | |
| Weather | 21 | 96 | 96 | (36696, 5175, 10444) | 10min | Weather |
| | | | 192 | (36600, 5079, 10348) | | |
| | | | 336 | (36456, 4935, 10204) | | |
| | | | 720 | (36072, 4551, 9820) | | |

---

[9]https://www.bgc-jena.mpg.de/wetter/

**Metric details.** We utilize the mean square error (MSE) and mean absolute error (MAE) for time series forecasting. The calculations of these metrics are:

$$\mathbf{MSE} = \Big(\sum_{i=0}^{L}(\mathbf{Y_i} - \hat{\mathbf{Y}_i})^2\Big)^{\frac{1}{2}}, \quad \mathbf{MAE} = \sum_{i=1}^{L}|\mathbf{Y_i} - \hat{\mathbf{Y}_i}|,$$

where $\mathbf{Y}, \hat{\mathbf{Y}} \in \mathbb{R}^{L \times C}$ are the ground-truth and the forecasting results with $L$ time points and $C$ dimensions of multivariate, respectively. $\mathbf{Y_i}$ means the $i$th future time point.

**Experiment details.** Since we only modify the loss function, whose configuration is detailed in the main text, all other training parameters, including learning rate, batch size, epochs, and model-specific hyperparameters, are left at their default settings for each respective method. The original code for these methods is fully open-sourced in their respective original publications (we summarize the URL links of these models used in our paper in Table 7), allowing for straightforward reproduction.

Table 7: The URL links of the models we used in this paper.

| Model | Backbone | URL Link |
|---|---|---|
| TimeMixer | MLP | https://github.com/kwuking/TimeMixer.git |
| TimesNet | MLP | https://github.com/thuml/TimesNet.git |
| Autoforemer | Transformer | https://github.com/thuml/Autoformer.git |
| Crossformer | Transformer | https://github.com/Thinklab-SJTU/Crossformer.git |
| iTransformer | Transformer | https://github.com/thuml/iTransformer.git |
| GPT2 | LLM (Transformer) | https://github.com/DAMO-DI-ML/NeurIPS2023-One-Fits-All.git |
| TimesFM | LLM (Transformer) | https://github.com/google-research/timesfm.git |
| Dlinear | MLP | https://github.com/cure-lab/LTSF-Linear.git |
| FEDformer | Transformer | https://github.com/MAZiqing/FEDformer.git |
| PatchTST | Transformer | https://github.com/yuqinie98/PatchTST.git |

# B MORE RESULTS OF MAIN EXPERIENTS

## B.1 MORE MODELS

In addition to the 3 models compared in the main text, we include 7 more models for a broader comparison. These include two MLP-based models: TimeMixer (Wang et al., 2024) and TimesNet (Wu et al., 2022); three Transformer-based models: Autoformer (Wu et al., 2021), Crossformer (Zhang & Yan, 2023), and iTransformer (Liu et al., 2023); and two recent LLM-based models: GPT2 (Zhou et al., 2023) and TimesFM (Das et al., 2023). The results are presented in Table 8.

Table 8: Multivariate time series forecasting results on more deep learning methods with/without hybrid loss framework.

| Models | Metrics | TimeMixer | TimesNet | Autoforemer | Crossformer | iTransformer | GPT2 | TimesFM | Dlinear | Dlinear (Hybrid Loss) | FEDformer | FEDformer (Hybrid Loss) | PatchTST | PatchTST (Hybrid Loss) |
|---|---|---|---|---|---|---|---|---|---|---|---|---|---|---|
| ETTh1 | MSE | 0.4512 | 0.4609 | 0.4738 | 0.5987 | 0.4570 | 0.4681 | 0.5406 | 0.4588 | 0.4579 | 0.4394 | **0.4380** | 0.4506 | 0.4502 |
| | MAE | 0.4405 | 0.4551 | 0.4733 | 0.5586 | 0.4492 | 0.4558 | 0.4519 | 0.4511 | 0.4581 | 0.4573 | 0.4411 | | **0.4402** |
| ETTh2 | MSE | 0.3849 | 0.4074 | 0.4258 | 0.5662 | 0.3837 | 0.3792 | **0.3127** | 0.4981 | 0.4974 | 0.4429 | 0.4417 | 0.3658 | 0.3639 |
| | MAE | 0.4061 | 0.4211 | 0.4447 | 0.5451 | 0.4069 | 0.4054 | **0.3748** | 0.4792 | 0.4785 | 0.4549 | 0.4539 | 0.3945 | 0.3929 |
| ETTm1 | MSE | 0.3908 | 0.4101 | 0.5502 | 0.5065 | 0.4076 | 0.3875 | 0.5240 | 0.4061 | 0.4060 | 0.4441 | 0.4424 | 0.3838 | **0.3813** |
| | MAE | 0.4023 | 0.4177 | 0.5024 | 0.5030 | 0.4118 | 0.4020 | 0.4577 | 0.4102 | 0.4102 | 0.4543 | 0.4535 | 0.3954 | **0.3943** |
| ETTm2 | MSE | **0.2767** | 0.2950 | 0.3251 | 1.5484 | 0.2922 | 0.2852 | 0.2852 | 0.3102 | 0.3100 | 0.3031 | 0.3021 | 0.2821 | 0.2790 |
| | MAE | **0.3232** | 0.3317 | 0.3637 | 0.7716 | 0.3358 | 0.3287 | 0.3586 | 0.3670 | 0.3667 | 0.3493 | 0.3480 | 0.3261 | 0.3247 |
| Electricity | MSE | 0.1818 | 0.1941 | 0.2370 | 0.3065 | 0.1756 | **0.1626** | 0.1860 | 0.2095 | 0.2093 | 0.2141 | 0.2224 | 0.1951 | 0.1955 |
| | MAE | 0.2722 | 0.2956 | 0.3436 | 0.3583 | 0.2666 | **0.2558** | 0.2667 | 0.2956 | 0.2955 | 0.3261 | 0.3334 | 0.2794 | 0.2796 |
| Exchange | MSE | 0.4356 | 0.4093 | 0.4901 | 0.9711 | 0.3642 | 0.3624 | **0.2313** | 0.3357 | 0.3307 | 0.5017 | 0.5201 | 0.3517 | 0.3531 |
| | MAE | 0.4298 | 0.4403 | 0.4929 | 0.7315 | 0.4069 | 0.4065 | **0.3328** | 0.3948 | 0.3947 | 0.4908 | 0.5025 | 0.3963 | 0.3966 |
| illness | MSE | 1.7500 | 2.2410 | 3.0330 | 3.7904 | 2.1360 | 1.9338 | 2.8652 | 2.3465 | 2.3452 | 2.7893 | 2.4759 | 1.6318 | **1.5197** |
| | MAE | 0.8706 | 0.9234 | 1.2053 | 1.2825 | 1.0075 | 0.9016 | 1.1173 | 1.0883 | 1.0892 | 1.1200 | 1.0974 | 0.8616 | **0.8279** |
| Weather | MSE | **0.2459** | 0.2588 | 0.3366 | 0.2638 | 0.2598 | 0.2548 | 0.2750 | 0.2670 | 0.2638 | 0.3128 | 0.3112 | 0.2598 | 0.2605 |
| | MAE | **0.2750** | 0.2857 | 0.3825 | 0.3229 | 0.2805 | 0.2780 | 0.2788 | 0.3174 | 0.3076 | 0.3609 | 0.3589 | 0.2816 | 0.2798 |

**Even with the increasing prevalence of LLM-based time series forecasting methods, our hybrid loss framework still enables existing models to achieve state-of-the-art performance in most cases.** The results in Table 8 demonstrate that, across 8 datasets, methods using our hybrid loss framework achieve state-of-the-art performance on 3 datasets, matching the number achieved by LLM-based methods and tying for the overall lead. This highlights the significant improvements provided by our hybrid loss framework for existing non-LLM methods and further suggests that there is still room for improvement in these methods.

### B.2 RESULTS OF EACH PREDICTION LENGTH

Here, we present the results for each prediction length in our main experiment.

Table 9: Multivariate time series forecasting results on deep learning methods with/without hybrid loss framework (prediction length is 96).

| Datasets | Models Loss | Dlinear Original | Hybrid Loss | FEDformer Original | Hybrid Loss | Patchtst Original | Hybrid Loss |
|---|---|---|---|---|---|---|---|
| ETTh1 | MSE | 0.3829 | **0.3779** | 0.3771 | **0.3770** | 0.3935 | **0.3924** |
| | MAE | **0.3959** | 0.3960 | 0.4185 | **0.4184** | 0.4080 | **0.4061** |
| ETTh2 | MSE | 0.3290 | **0.3279** | 0.3508 | **0.3481** | 0.2938 | **0.2927** |
| | MAE | 0.3804 | **0.3795** | 0.3918 | **0.3902** | 0.3427 | **0.3415** |
| ETTm1 | MSE | 0.3458 | **0.3457** | 0.3669 | **0.3628** | 0.3211 | **0.3183** |
| | MAE | **0.3737** | **0.3737** | 0.4122 | **0.4097** | 0.3596 | **0.3572** |
| ETTm2 | MSE | **0.1869** | **0.1869** | 0.1918 | **0.1908** | 0.1776 | **0.1758** |
| | MAE | 0.2811 | **0.2810** | 0.2812 | **0.2801** | 0.2599 | **0.2586** |
| Electricity | MSE | 0.1946 | **0.1944** | 0.1884 | 0.1950 | 0.1718 | **0.1664** |
| | MAE | 0.2774 | **0.2773** | 0.3036 | 0.3095 | 0.2573 | **0.2332** |
| Exchange | MSE | 0.0782 | **0.0779** | 0.1447 | 0.16653 | 0.0806 | **0.0805** |
| | MAE | 0.1985 | **0.1977** | 0.2736 | 0.2942 | 0.1973 | **0.1965** |
| illness | MSE | 2.2795 | **2.2794** | 3.2211 | **2.7505** | 1.7609 | **1.4334** |
| | MAE | **1.0601** | 1.0622 | 1.2420 | **1.1599** | 0.9018 | **0.8020** |
| Weather | MSE | 0.1969 | **0.1968** | 0.2231 | 0.2232 | 0.1816 | **0.1769** |
| | MAE | 0.2551 | **0.2550** | 0.3051 | 0.3061 | 0.2219 | **0.2157** |

## C MORE RESULTS OF ABLATION STUDY

Due to the high computational cost of the Fourier transform in FEDformer, the second ablation study as described in the main text is conducted only on the ETTh1 and ETTh2 datasets for FEDformer. Results are shown in Table 13. The results further corroborate the conclusions presented in the main paper, which confirm that **uniform initial weights (set to 0.5) is an effective initialization strategy, allowing the model to subsequently and efficiently adjust the individual loss weights**.

## D MORE SHOWCASES

This section presents additional cases using the original overall loss and our hybrid loss framework, as illustrated in Figure 3, Figure 4, and Figure 5. Notably, we show the results of the forecasting part with the settings of the input length 96 and prediction length 96 in the main text. Therefore, we show the results of the forecasting part with the settings of the input length 96 and prediction length {192, 336, 720} here, respectively. These results further support our conclusions from the main text: **the original overall loss may lead to large errors in individual sub-series, further hindering overall forecasting performance, and our hybrid loss framework effectively mitigates this issue.**

Table 10: Multivariate time series forecasting results on deep learning methods with/without hybrid loss framework (prediction length is 192).

| Datasets | Models | Dlinear | | FEDformer | | Patchtst | |
|---|---|---|---|---|---|---|---|
| | Loss | Original | Hybrid Loss | Original | Hybrid Loss | Original | Hybrid Loss |
| ETTh1 | MSE | **0.4327** | **0.4327** | 0.4200 | **0.4198** | **0.4453** | 0.4464 |
| | MAE | **0.4258** | **0.4258** | 0.4441 | **0.4439** | 0.4342 | **0.4338** |
| ETTh2 | MSE | **0.4313** | 0.4333 | 0.4420 | **0.4407** | 0.3769 | **0.3744** |
| | MAE | **0.4432** | 0.4446 | 0.4498 | **0.4482** | 0.3930 | **0.3913** |
| ETTm1 | MSE | 0.3826 | **0.3825** | 0.4360 | **0.4345** | 0.3652 | **0.3625** |
| | MAE | 0.3929 | **0.3928** | 0.4465 | **0.4453** | **0.3820** | 0.3828 |
| ETTm2 | MSE | 0.2720 | **0.2712** | 0.2637 | **0.2636** | 0.2487 | **0.2408** |
| | MAE | 0.3486 | **0.3477** | 0.3255 | **0.3252** | 0.3064 | **0.3023** |
| Electricity | MSE | **0.1939** | **0.1939** | **0.1964** | 0.2023 | **0.1789** | 0.1824 |
| | MAE | **0.2804** | **0.2804** | **0.3109** | 0.3156 | **0.2647** | 0.2685 |
| Exchange | MSE | **0.1559** | 0.1562 | **0.2648** | 0.2706 | 0.1710 | **0.1704** |
| | MAE | **0.2921** | 0.2926 | **0.3745** | 0.3812 | 0.2931 | **0.2920** |
| illness | MSE | 2.2350 | **2.2323** | 2.5884 | **2.3293** | **1.4001** | 1.5344 |
| | MAE | **1.0580** | 1.0586 | 1.1204 | **1.0973** | 0.8616 | **0.8279** |
| Weather | MSE | 0.2392 | **0.2265** | 0.2847 | **0.2782** | **0.2271** | 0.2275 |
| | MAE | 0.2971 | **0.2582** | 0.3547 | **0.3454** | 0.2601 | **0.2582** |

Table 11: Multivariate time series forecasting results on deep learning methods with/without hybrid loss framework (prediction length is 336).

| Datasets | Models | Dlinear | | FEDformer | | Patchtst | |
|---|---|---|---|---|---|---|---|
| | Loss | Original | Hybrid Loss | Original | Hybrid Loss | Original | Hybrid Loss |
| ETTh1 | MSE | 0.4913 | **0.4912** | 0.4581 | **0.4577** | **0.4838** | 0.4843 |
| | MAE | 0.4673 | **0.4671** | 0.4664 | **0.4659** | 0.4515 | **0.4511** |
| ETTh2 | MSE | **0.4586** | 0.4604 | 0.4985 | **0.4982** | 0.3806 | **0.3800** |
| | MAE | **0.4618** | 0.4633 | 0.4905 | **0.4902** | 0.4089 | 0.4091 |
| ETTm1 | MSE | **0.4165** | **0.4165** | 0.4666 | **0.4659** | 0.3933 | **0.3910** |
| | MAE | **0.4175** | **0.4175** | **0.4699** | 0.4702 | **0.4039** | 0.4052 |
| ETTm2 | MSE | 0.3434 | **0.3433** | 0.3306 | **0.3267** | 0.3033 | **0.3027** |
| | MAE | **0.3945** | **0.3945** | 0.3674 | **0.3637** | **0.3411** | 0.3423 |
| Electricity | MSE | **0.2069** | **0.2069** | **0.2076** | 0.2289 | **0.1946** | 0.1975 |
| | MAE | 0.2963 | **0.2963** | **0.3231** | 0.3420 | **0.2811** | 0.2893 |
| Exchange | MSE | 0.3269 | **0.3071** | **0.4437** | 0.4740 | **0.3188** | 0.3202 |
| | MAE | **0.4192** | 0.4194 | **0.4923** | 0.5049 | **0.4070** | 0.4078 |
| illness | MSE | 2.2983 | **2.2925** | 2.5682 | **2.3153** | 1.6891 | **1.5918** |
| | MAE | 1.0788 | **1.0773** | 1.0591 | **1.0161** | **0.8431** | 0.8649 |
| Weather | MSE | 0.2835 | **0.2834** | 0.3277 | **0.3112** | **0.2792** | 0.2817 |
| | MAE | 0.3324 | **0.3323** | 0.3651 | **0.3589** | 0.2983 | **0.2981** |

Table 12: Multivariate time series forecasting results on deep learning methods with/without hybrid loss framework (prediction length is 720).

| Datasets | Models Loss | Dlinear Original | Dlinear Hybrid Loss | FEDformer Original | FEDformer Hybrid Loss | Patchtst Original | Patchtst Hybrid Loss |
|---|---|---|---|---|---|---|---|
| ETTh1 | MSE | **0.5284** | 0.5296 | **0.5022** | 0.5053 | 0.4798 | **0.4778** |
|  | MAE | **0.5185** | 0.5193 | **0.5032** | 0.5050 | 0.4707 | **0.4697** |
| ETTh2 | MSE | 0.7736 | **0.7719** | 0.4804 | **0.4798** | 0.4118 | **0.4102** |
|  | MAE | 0.6313 | **0.6306** | 0.4873 | **0.4870** | 0.4334 | **0.4325** |
| ETTm1 | MSE | 0.4794 | **0.4792** | 0.5068 | **0.5065** | 0.4556 | **0.4535** |
|  | MAE | 0.4567 | **0.4566** | 0.4887 | **0.4887** | **0.4359** | 0.4401 |
| ETTm2 | MSE | 0.4385 | **0.4383** | 0.4263 | **0.4252** | 0.3986 | **0.3968** |
|  | MAE | 0.4439 | **0.4437** | 0.4229 | **0.4221** | 0.3969 | **0.3954** |
| Electricity | MSE | **0.2425** | **0.2425** | 0.2639 | **0.2634** | **0.2349** | 0.2357 |
|  | MAE | 0.3283 | **0.3283** | **0.3669** | 0.3664 | **0.3146** | 0.3274 |
| Exchange | MSE | 0.7816 | **0.7815** | **1.1535** | 1.1708 | **0.8363** | 0.8411 |
|  | MAE | 0.6692 | **0.6690** | **0.8227** | 0.8304 | **0.6879** | 0.6898 |
| illness | MSE | **2.5735** | 2.5765 | 2.7804 | **2.5085** | 1.6775 | **1.5200** |
|  | MAE | **1.1578** | 1.1591 | 1.1323 | **1.1163** | 0.8754 | **0.8052** |
| Weather | MSE | 0.3484 | **0.3483** | **0.3721** | 0.3848 | **0.3514** | 0.3557 |
|  | MAE | 0.3849 | **0.3848** | 0.4187 | **0.3589** | **0.3461** | 0.3473 |

Table 13: The ablation study of multivariate time series forecasting results on our hybrid loss framework with different initial weights. As $w_1 + w_2 = 1$ and $\alpha + \beta = 1$, we only specify the initial values of $w_1$ and $\alpha$ in the table.

| Models | $w_1$ | $\alpha$ | ETTh1 MSE | ETTh1 MAE | ETTh2 MSE | ETTh2 MAE | ETTm1 MSE | ETTm1 MAE | ETTm2 MSE | ETTm2 MAE |
|---|---|---|---|---|---|---|---|---|---|---|
| Dlinear | 0.1 | 0.1 | 0.4596 | 0.4524 | 0.4939 | 0.4779 | 0.4091 | 0.4142 | 0.3064 | 0.3638 |
|  |  | 0.9 | 0.4610 | 0.4541 | 0.4978 | 0.4791 | 0.4047 | 0.4097 | 0.3091 | 0.3658 |
|  | 0.5 | 0.5 | 0.4579 | 0.4511 | 0.4974 | 0.4785 | 0.4060 | 0.4102 | 0.3100 | 0.3667 |
|  | 0.9 | 0.1 | 0.4589 | 0.4519 | 0.4982 | 0.4793 | 0.4050 | 0.4103 | 0.3102 | 0.3670 |
|  |  | 0.9 | 0.4589 | 0.4519 | 0.4983 | 0.4793 | 0.4050 | 0.4103 | 0.3102 | 0.3670 |
| Patchtst | 0.1 | 0.1 | 0.4554 | 0.4451 | 0.3651 | 0.3940 | 0.3836 | 0.3975 | 0.2802 | 0.3200 |
|  |  | 0.9 | 0.4518 | 0.4412 | 0.3650 | 0.3937 | 0.3828 | 0.3973 | 0.2800 | 0.3254 |
|  | 0.5 | 0.5 | 0.4502 | 0.4402 | 0.3639 | 0.3929 | 0.3813 | 0.3943 | 0.2790 | 0.3247 |
|  | 0.9 | 0.1 | 0.4493 | 0.4393 | 0.3642 | 0.3935 | 0.3821 | 0.3960 | 0.2792 | 0.3247 |
|  |  | 0.9 | 0.4492 | 0.4391 | 0.3642 | 0.3934 | 0.3821 | 0.3935 | 0.2792 | 0.3248 |
| FEDformer | 0.1 | 0.1 | 0.4428 | 0.4601 | 0.4432 | 0.4549 | — | — | — | — |
|  |  | 0.9 | 0.4421 | 0.4596 | 0.4441 | 0.4556 | — | — | — | — |
|  | 0.5 | 0.5 | 0.4380 | 0.4573 | 0.4417 | 0.4539 | — | — | — | — |
|  | 0.9 | 0.1 | 0.4394 | 0.4581 | 0.4430 | 0.4548 | — | — | — | — |
|  |  | 0.9 | 0.4394 | 0.4581 | 0.4430 | 0.4549 | — | — | — | — |

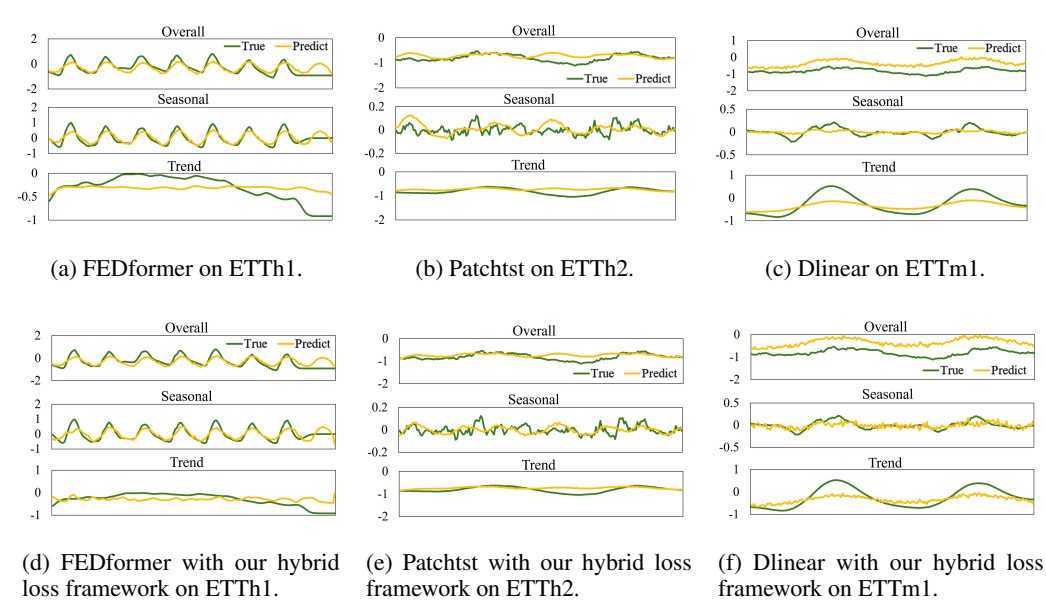

(a) FEDformer on ETTh1.  (b) Patchtst on ETTh2.  (c) Dlinear on ETTm1.

(d) FEDformer with our hybrid loss framework on ETTh1.  (e) Patchtst with our hybrid loss framework on ETTh2.  (f) Dlinear with our hybrid loss framework on ETTm1.

Figure 3: The case study of time series forecasting. The results show the prediction-length-192 part (input length is 96) for different methods on different datasets. Each sub figure presents the single-variate (last variate) overall forecasting part and the forecasting part of the individual sub-series.

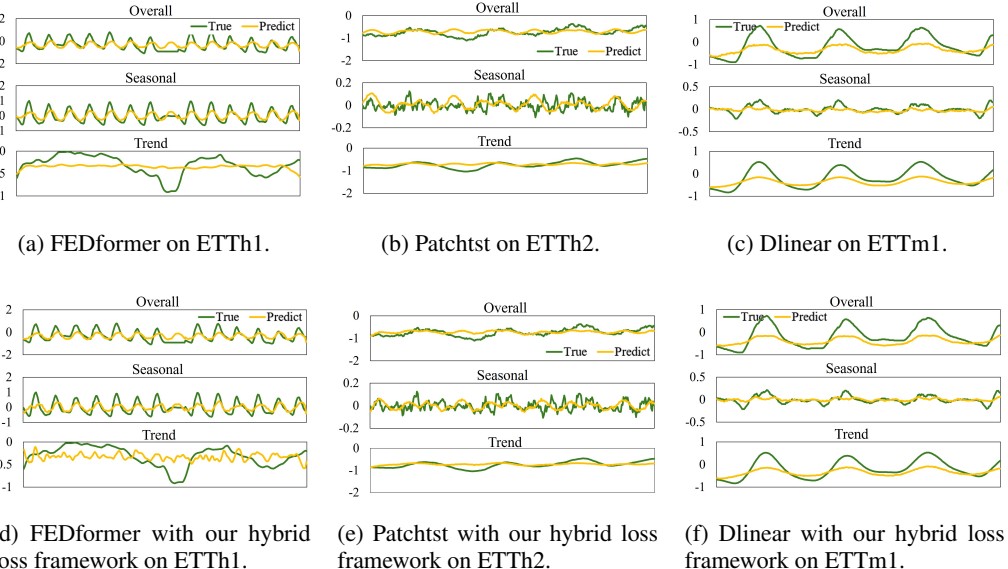

(a) FEDformer on ETTh1.  (b) Patchtst on ETTh2.  (c) Dlinear on ETTm1.

(d) FEDformer with our hybrid loss framework on ETTh1.  (e) Patchtst with our hybrid loss framework on ETTh2.  (f) Dlinear with our hybrid loss framework on ETTm1.

Figure 4: The case study of time series forecasting. The results show the prediction-length-336 part (input length is 96) for different methods on different datasets. Each sub figure presents the single-variate (last variate) overall forecasting part and the forecasting part of the individual sub-series.

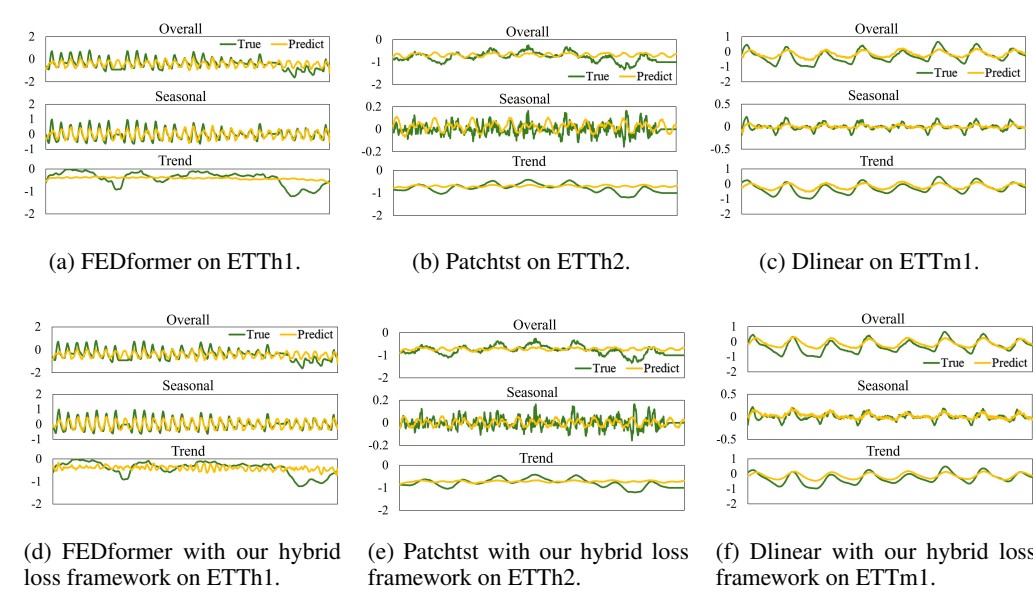

(a) FEDformer on ETTh1.

(b) Patchtst on ETTh2.

(c) Dlinear on ETTm1.

(d) FEDformer with our hybrid loss framework on ETTh1.

(e) Patchtst with our hybrid loss framework on ETTh2.

(f) Dlinear with our hybrid loss framework on ETTm1.

Figure 5: The case study of time series forecasting. The results show the prediction-length-720 part (input length is 96) for different methods on different datasets. Each sub figure presents the single-variate (last variate) overall forecasting part and the forecasting part of the individual sub-series.

