# OpenReview forum: "A Hybrid Loss Framework for Decomposition-based Time Series Forecasting Methods: Balancing Global and Component Errors"
_ICLR.cc/2025/Conference — ICLR 2025 Conference Withdrawn Submission_

### Official Review · Reviewer_zpzN · 2024-11-01

**Soundness:** 2
**Presentation:** 2
**Contribution:** 2
**Rating:** 3
**Confidence:** 5

**Summary:**

The authors propose a novel loss function for sliding window-based time series decomposition models. They employ a dual weighted minmax loss to better account for errors made on both the global prediction (sum of the prediction of both decomposition components) and on the prediction of each component of the decomposition. During training, the model assigns varying importance to the losses through learned weights.

The model is evaluated against a set of baselines on eight well-known time series datasets.

**Strengths:**

- Legitimate question
- Interesting proposal

**Weaknesses:**

- Missing important decomposition SOTA
- Improvements are limited
- Preliminary experiments are not adequate to justify the problem/question appropriately
- Thorough proof-read is required

**Questions:**

## Terminology

### T1
> […] high-frequency feature (detailed changes) and low frequency feature (overall trend changes), respectively (Faltermeier et al., 2010).

I appreciate the effort in introducing moving average decomposition as it should be. However, authors should have kept using correct descriptions of components (moving average component and remainder component), as, contrary to the literature, the first term is not necessarily “seasonal” and the second is not necessarily a “trend”. It is more appropriate to call the latter a residual or remainder component as it is defined as $x\_r = x - x\_{ma}$ in most literature (especially models inherited from Autoformer). The variations in your plots (cf. Figure 3, 4, and 5) show that this residual may sometimes hold seasonal information and other times just be a trend or noise. These variations could depend on the dataset, the input length, and the moving average window.

### T2
> […] global and component errors.

Would it not be more appropriate, looking at the paper, to talk about “global and components losses”?

### T3
> Does optimizing this overall loss guarantee that the features of each sub-series are equally well-learned? Or, could an optimal overall loss fail to optimize the performance of the decomposition-based deep learning model?

Unfortunately, I don’t feel that the paper really answers these questions appropriately.

## Main Issues
Even though mentioned in the limitation section, SOTA decomposition models are not presented in the related works sections or the paper, limiting the results presented here to sliding window-based decomposition only.

Especially, LD has shown to outperform sliding window-based decomposition models.
- https://arxiv.org/abs/2402.12694
- https://arxiv.org/pdf/2405.10877
- https://arxiv.org/pdf/2308.13386


## Questions

### Q1
Why are the plots only for ETTh1, ETTh2, and ETTm1, even in the appendix? To allow readers to draw their own conclusions, it is better to provide plots for all datasets. It is also important for reproducibility to inform which channel is plotted. For ETTx, are we looking at Oil Temperature? As this is a multivariate TS Forecasting, it might be more appropriate to show more and specify which channels to avoid ending up on a channel that will favor one model over the others.

### Q2
> Therefore, this further confirms that the overall loss may not effectively optimize for the sub-series that contribute significantly to the overall forecasting.

Would it not be more explanatory to provide a box plot of the loss for all samples, $loss(x_g, \hat{x_g})$, $loss(x_{ma}, \hat{x_{ma}})$, and $loss(x_r, \hat{x_r})$? [here $ma$ is your "seasonal" component and $r$ is your "trend" component, $x\_g = x\_{ma} + x\_r$]
Because your plot is only the comparison of one sample considering one channel, so it is difficult to generalize. In fact, for ETTm1, maybe the sample you selected does not have good performance on seasonal, but another one would have. And vice-versa for ETThi.

### Q3
> Our hybrid loss framework effectively improves the final performance of existing methods across a wide range of datasets.

> Our hybrid loss framework significantly enhances the forecasting performance of individual sub-series, particularly the Trend sub-series.

Are the results in Table 2 really significant? Improvement seems substantial. And differences in Table 3 seem also very substantial. For instance, improvement from $0.0888$ to $0.0819$ is not what I would call significant.

### Q4
> Comparing Figure 1 with Figure 2, we observe a clear visual improvement in the forecasting of sub-series, particularly the Trend sub-series in Figure 2 (a) and (b).

I disagree with this claim. For Figure 2(a), it might be just because the y-axis is different (-1 to 0 in Figure 1(a), -2 to 0 in Figure 2(a)). Regarding (b), for me, there are no differences. Especially, comparing Figure 1, which is the main insight of the “issue”, with Figure 2, the performance with Hybrid loss does not seem to improve the decomposition. Indeed, all the claims from the last paragraph of Section 2 could apply to Figure 2, which would imply that the “problem” is not solved.

### Q5
> DLinear on ETTm1 also shows a visually perceptible improvement in overall performance, despite some residual high-frequency errors in the Seasonal sub-series.

In my opinion, there are no differences.

Please provide a similar plot that includes both with and without Hybrid loss, to visually check the difference (or plot the difference between with HL and without).

### Q6 - Ablation study:
I would like to see the performance of a version with $0.33LossG + 0.33LossS + 0.33LossT$, to better account for the role of $w\_1$ and $\alpha$.
Something not based on a dual minmax loss.

### Q7
The “loss” in Table 2, is the metric to evaluate performance or the loss function used during training? Make it clearer in the paper.

### Q8
> This demonstrates that the dynamic focus on sub-series losses introduced by our hybrid loss framework is indeed effective and ultimately leads to improved overall performance.

Again, I have the feeling that the difference is a matter of dataset rather than loss. Especially, as Hybrid loss is making results worse for some datasets and improvement is not so significant for the others.
But how many times have the experiments been run? What is the standard deviation of performances? As these could also be a matter of seed used.

### Q9
We miss a study on the impact of lambda on the performance.


## Code
Why define $w\_2$ and $\beta$? It would be simpler to use $1 - w\_1$ and $1 - \alpha$. It would notably avoid having cases where $w\_1 + w\_2$ is not equal to $1$ or $\alpha + \beta$ is not equal to $1$, as such a property is not in your code, so nothing prevents the model from doing so.

Reproducibility: code as-is is not usable, this is not the code used for the results as there are typos (lambda).

## Limitations

### L1
> These methods all employ sliding window-based time series decomposition […]

To the best of my knowledge, except for FEDformer, which is slightly different as frequency transformation is applied, the models on which authors are applying their hybrid loss use the same decomposition mechanism (inherited from Autoformer). As a result, it raises the question that this hybrid loss might be “beneficial” for this kind of decomposition. It is of crucial importance to include other models using different decomposition such as the one mentioned above in order to really assess the efficiency of the proposed hybrid loss. Authors mentioned this in limitation, but, in my opinion, it should have been tackled directly with this submission. If authors do not include other decomposition techniques, then it needs to be clear in the paper that this hybrid loss is defined and tested for sliding window-based decomposition **only**, without proper experiment it is not possible to generalize to any decomposition methods.

### L2
> […] significant discrepancies in forecasting performance across individual sub-series, when trained under the overall loss, may contribute to mainly inaccuracies in the final combined forecasting.

The preliminary experiment results are, in my opinion, limited as it targets the same nature of dataset. ETTx all represent the same type of data, so the chance of finding the same behavior is high. Especially, as ETThi is basically an aggregation on an hourly level of ETTmi. To really be able to conclude something from this experiment, it would be more appropriate to test different datasets.

In addition, the efficiency of the decomposition strongly depends on the window size, it would be interesting to test if these discrepancies are due to the global loss or the usage of a fixed moving average setting that might not be optimal for all datasets.

Furthermore, following your argument (on Figure (c) DLinear is better on predicting “trend” compared to “seasonal” as shown in the results in Table 1), then, for FEDformer and ETTh1, the model should perform worse on “Seasonal” than “trend”, but this is not the case in Figure 1. Such an observation led me to think that focusing only on results from Table 1 to conclude that global loss is not appropriate for decomposition is not adapted and more experiment would have been required.

## Revision
 * Lines 383 & 384: In Latex, opening quotes should be done with `` (two single backquotes) not “
 * Missing SOTA model PathFormer for Table 8

## Proof-Read
To cite but a few:
- Uniformize the names of models, sometimes FEDformer, Fedformer PatchTST, Patchtst, etc.
- “[...] series decomposition(Cleveland et al., 1990; [...]” -> missing space before citation
- “[…] combining the global (the overall loss) and component error […]” -> errors
- “This indicates that a overall loss may not ensure consistent [...]” -> an
- “[…] a overall loss indeed introduces bias […]” -> an
- “[…] while also focusing on and dynamically balancing the forecasting of sub-series with larger errors […]” -> "on dynamically"
- “[…] the model also dynamically attends to the overall sub-series loss.” "attends"? Do authors mean “pay attention”?
- “We employe the original loss function […]” -> “employ”?
- In table 4, “Componet” -> “Component”

---

### Official Review · Reviewer_dLRR · 2024-11-03

**Soundness:** 2
**Presentation:** 2
**Contribution:** 2
**Rating:** 3
**Confidence:** 3

**Summary:**

This paper presents a hybrid loss framework for decomposition-based time series forecasting, balancing overall forecasting accuracy with component-specific errors (e.g., trend and seasonal). The key contribution is a dual min-max optimization approach that dynamically adjusts the weighting between global and component losses. The framework is empirically validated on multiple datasets, showing improved performance for methods like DLinear, FEDformer, and PatchTST.

**Strengths:**

- The dual min-max optimization addresses a specific challenge in decomposition-based forecasting by incorporating sub-series accuracy, which could inform future frameworks in similar areas.

- The paper is well-organized, with a logical flow from problem identification to empirical validation. The equations are systematically presented, and the framework is thoroughly explained.

- The experiments are extensive, covering diverse datasets and evaluating the framework on established models. The use of ablation studies and detailed hyperparameter settings also improves reproducibility.

**Weaknesses:**

- The connection between the two min-max problems is not fully explained. The paper could benefit from a deeper theoretical analysis of convergence properties, bounds on component-wise performance, and an exploration of optimality conditions.

- The paper primarily compares decomposition-based models (DLinear, FEDformer, PatchTST) but would be more compelling if it included non-decomposition models as baselines to showcase the framework’s versatility.

- The dual optimization framework potentially introduces computational overhead. An analysis of the computational efficiency of this approach compared to traditional end-to-end loss functions would provide clarity on the framework’s scalability.

**Questions:**

- What theoretical guarantees can be provided about the convergence of the dual min-max optimization framework?

- How does the performance vary across different initializations of $\omega_1$, $\omega_2$, $\alpha$, and $\beta$? Are there recommended initializations?

- The Dual Min-Max Optimization Setup lacks theoretical insights into why balancing $\omega_1$, and $\omega_2$, in this manner effectively improves component-wise learning and performance. Clarification on why this objective is expected to optimize both global and component forecasts would be helpful.

- The paper does not explain why the trade-off between seasonal and trend components is generalized across datasets with potentially different characteristics. This optimization assumes equal importance for all datasets, which may not hold in practice. How does the model adapt to varying sub-series structures across datasets?

- The mirror descent technique from DRO is used without explaining why it’s suitable for this problem. Why not use more straightforward optimization methods for updating the weights? It is unclear if the mirror descent method guarantees convergence in this setting. A theoretical analysis of the convergence properties or stability of the update rules would make this section more robust.

- What would the implications be of using different decomposition methods (e.g., Fourier or wavelet transforms) with this framework? Could it be adapted for models without sliding-window decomposition? How about the non-decomposition models?

---

### Official Review · Reviewer_CD45 · 2024-11-03

**Soundness:** 2
**Presentation:** 2
**Contribution:** 2
**Rating:** 1
**Confidence:** 5

**Summary:**

The authors present a study focused on the importance of monitoring the performance of individual time series components, in contrast to established methodologies that emphasize overall performance. The experiments include methods recently published in the time series literature. In addition to addressing the lack of studies on this specific issue, the authors propose an algorithm that combines component-specific and overall losses to enhance the accuracy of time series predictions.

**Strengths:**

The manuscript is well-written and well-structured, with minor grammatical errors that do not affect overall comprehension. The investigation proposed by the authors is promising and has the potential to advance the state of the art.

**Weaknesses:**

Although I consider the proposed study creative and relevant for advancing the current state of the art, several points require further investigation to effectively demonstrate the advantages of the authors' approach:

1) The results are currently limited. Statistical tests were not deeply explored to highlight significant differences between the proposed method and baselines.

2) Running the experiments only once on the time series may capture differences that reflect local behavior. Applying a windowed approach to the experiments could help reduce potential biases.

3) The components analyzed appear to be additive, and while the authors have explored various combinations of these components, broader exploration may be beneficial.

4) Beyond trend and seasonality, other components, such as stochastic ones, should also be examined.

5) The authors should investigate causal relationships between components and overall performance to avoid drawing conclusions based on spurious correlations.

**Questions:**

1) The authors change datasets across their experiments without clearly justifying these choices. For example, four datasets are shown in Table 1, eight in Table 2, and four in Table 3. Why didn't the authors use a consistent number across these discussions? Additionally, in Section 2, the experimental settings do not mention the new datasets introduced in Table 2. What are the similarities and differences between the datasets? What types of behavior do they allow for investigation? The authors are encouraged to provide a clear justification for their dataset selections across the experiments. Additionally, they should explain any differences between the datasets used and describe the key characteristics of each dataset that make them appropriate for this study.

2) The reported performance improvements involve differences as small as the fourth decimal place. Is this difference meaningful, or could it be a rounding artifact? The authors are advised to conduct and report statistical significance tests on the observed performance differences. Additionally, they should consider explaining why even small improvements might be meaningful in this context. Moreover, I recommend using a sliding-window strategy to collect more robust results from each time series.

3) Why did the authors decompose time series into seasonal and trend components? Please discuss potential benefits or drawbacks of exploring alternative techniques to extract further components, like Empirical Mode Decomposition.

4) In the list of contributions (Introduction), the final point is simply a brief discussion of results. This does not qualify as a "contribution."

5) The authors chose MAE and MSE as evaluation metrics, but other measures, like DTW or MAPE, could provide more relevant insights aligned with the study’s goals. The choice of metrics should be justified.

6) The "poor" result in Figure 1(c) may be explained by residual components, such as stochastic influences, that can impact overall performance if not properly modeled. Did the authors examine the influence of the stochastic component?

7) Figures 1 and 2 represent only three datasets. Why was this subset chosen?

8) The authors claim, "Comparing Figure 1 with Figure 2, we observe a clear visual improvement in the forecasting of sub-series." This conclusion seems subjective, particularly for Figure 2(b).

9) Many results in Table 2 are repeated in Table 3, with the only difference being the additional datasets in Table 2, which are not discussed in detail.

10) The conclusion section mainly summarizes the manuscript without drawing specific conclusions about the authors' hypothesis.

11) Has the manuscript been proofread? Minor errors need correction, such as "the Equation 1" and "the Equation 2."

12) The min-max approach requires more explanation. Do the authors intend to minimize in terms of θ (DNN parameters)? Is this strictly related to DNN training? The authors should provide a more comprehensive explanation of their min-max approach in Section 3, clarifying the optimization process, the role of DNN parameters, and how the approach is expected to improve results. It would also be helpful to provide a step-by-step breakdown of the algorithm to enhance reader understanding.

13) Is the error observed in each component influenced by its magnitude? Would a slight trend result in a similar error proportion? I recommend conducting and reporting an analysis of how component magnitude affects observed errors and discussing the implications of this relationship for the proposed method. This could include examining cases with slight trends or weak seasonality to assess how the approach performs in these scenarios.

---

### Official Review · Reviewer_K1dR · 2024-11-04

**Soundness:** 2
**Presentation:** 2
**Contribution:** 1
**Rating:** 1
**Confidence:** 5

**Summary:**

This study explores the impact of overall loss functions in time series forecasting, particularly for methods that decompose series into sub-series for separate processing. It questions whether using a single, overall loss function effectively prioritizes important sub-series, potentially affecting forecasting accuracy. The findings indicate that an overall loss can introduce bias, reducing the model’s focus on critical sub-series and limiting performance. To address this, the authors propose a hybrid loss framework that combines global and sub-series-specific losses, using a dual min-max algorithm to dynamically adjust the weights. This approach enhances model performance by focusing on critical sub-series without altering model architectures, yielding a 0.5-2% improvement on multiple datasets.

**Strengths:**

The paper addresses the issue of loss imbalance in forecasting methods that decompose a time series into sub-series, which can improve forecast accuracy in many applications. Overall, the paper is clear and easy to read.

**Weaknesses:**

* The authors focus on a very specific dataset, raising concerns about the generalizability of the findings. It is unclear if this phenomenon exists in other datasets. Additional datasets, such as the Monash Time Series Forecasting Archive (https://arxiv.org/abs/2105.06643), should be considered.
* The paper primarily addresses point forecasting, whereas probabilistic forecasting—which captures uncertainty more effectively—is generally considered a more relevant and important problem in the field.
* The results in Tables 2 and 3 indicate only marginal performance differences, often in the third or fourth decimal place.
* It is unclear if a simpler weighting scheme might suffice instead of the proposed distributionally robust optimization, especially considering the increased computational demands of the latter. The authors considered a fixed weighting scheme where all weights are set to 0.5, which is not ideal given the scale imbalance.

**Questions:**

See weaknesses.

---

### Note · Authors · 2024-11-18

**Comment:**

Dear Conference Organizing Committee and Reviewers,

Thank you for your time and valuable feedback on our paper. After carefully revisiting the specific comments provided by the reviewers, we have decided to revise the content of the article based on some of the insightful suggestions. Moreover, after further consideration, we have decided to retract the paper from submission.

We sincerely appreciate the time and effort the reviewers have invested in evaluating our work and are grateful for their thoughtful contributions.

**Withdrawal Confirmation:**

I have read and agree with the venue's withdrawal policy on behalf of myself and my co-authors.